# Cryopreservation of Cyanobacteria and Eukaryotic Microalgae Using Exopolysaccharide Extracted from a Glacier Bacterium

**DOI:** 10.3390/microorganisms9020395

**Published:** 2021-02-15

**Authors:** Pervaiz Ali, Daniel Fucich, Aamer Ali Shah, Fariha Hasan, Feng Chen

**Affiliations:** 1Institute of Marine and Environmental Technology, University of Maryland Center for Environmental Science, Baltimore, MD 21613, USA; pali@umces.edu (P.A.); dfucich@umces.edu (D.F.); 2Applied Environmental and Geomicrobiology Laboratory, Department of Microbiology, Quaid-i-Azam University, Islamabad 15320, Pakistan; alishah@qau.edu.pk (A.A.S.); farihahasan@yahoo.com (F.H.)

**Keywords:** psychrophilic bacteria, exopolysaccharide (EPS), cryopreservation, cyanobacteria, microalgae

## Abstract

Exopolysaccharide (EPS) has been known to be a good cryoprotective agent for bacteria, but it has not been tested for cyanobacteria and eukaryotic microalgae. In this study, we used EPS extracted from a glacier bacterium as a cryoprotective agent for the cryopreservation of three unicellular cyanobacteria and two eukaryotic microalgae. Different concentrations of EPS (10%, 15%, and 20%) were tested, and the highest concentration (20%) of EPS yielded the best growth recovery for the algal strains we tested. We also compared EPS with 5% dimethyl sulfoxide (DMSO) and 10% glycerol for the cryopreservation recovery. The growth recovery for the microalgal strains after nine months of cryopreservation was better than 5% DMSO, a well-known cryoprotectant for microalgae. A poor recovery was recorded for all the tested strains with 10% glycerol as a cryoprotective agent. The patterns of growth recovery for most of these strains were similar after 5 days, 15 days, and 9 months of cryopreservation. Unlike common cryopreservants such as DMSO or methanol, which are hazardous materials, EPS is safe to handle. We demonstrate that the EPS from a psychrotrophic bacterium helped in the long-term cryopreservation of cyanobacteria and microalgae, and it has the potential to be used as natural cryoprotective agent for other cells.

## 1. Introduction

Many eukaryotic microalgae and cyanobacteria have been isolated from natural environments. These microalgae contain many specific properties that have research and commercial value. They are cultivated and maintained in different laboratories worldwide, and many algal cultures have been deposited in culture collection centers. There is a need to preserve the cultures for longer time while ensuring viability, purity, and genetic stability. Cryopreservation is the maintenance of the biological samples in a state of ‘suspended animation’ at low temperatures [1]. Though cryopreservation (typically at −80 °C) has been successfully used in bacteria and cyanobacteria [2,3], it has not been routinely used to maintain complex eukaryotic microalgae [4]. Some photoautotrophic organisms have been maintained in laboratories through serial sub-culturing [5]. However, this method of culture maintenance has inherent disadvantages, including culture contamination (either bacterial or cross contamination with other strains), time- and labor-intensive processes, and expensive resources when it involves large culture collections [6].

Cryopreservation is an important technique for the long-term preservation of cells, tissues, and organs. However, the ultra-low temperature can cause the formation of ice crystals in the cellular cytoplasm in absence of a suitable cryoprotective agent, resulting in lysis the cell membrane [7]. Cryoprotective agents (CPAs) are added to culture media to protect cells from cryo-injury by lowering the freezing point of water and inhibiting ice crystal formation in the suspension medium and cell interior [8]. Dimethyl sulfoxide (DMSO), methanol, and glycerol are commonly used intracellular cryoprotectants, while sucrose and large polymers provide extracellular protection because these polymers cannot permeate the cell membrane [7]. CPAs, at higher concentrations, more effectively prevent ice formation in cells, tissues, and organs during cryopreservation. However, CPAs become more toxic at higher concentrations and impede the viabilities of the cryopreserved materials [9]. Successful cryopreservation should therefore involve strategies to eliminate ice crystal formation while minimizing the risk of CPA toxicity [10]. The various factors affecting successful cryopreservation depend on the types and concentrations of CPAs, the storage time, and the temperature. The method used for cryopreservation is paramount, as many strains cannot withstand deleterious steps such as pre-cooling and post-thawing. Thus, looking for alternative CPAs with less toxicity and improving protocols to avoid cryo-injury can help in the successful cryopreservation of cells.

The use of glycerol and DMSO, which are regarded as universally useful CPAs for most samples, actually marked the beginning of modern cryopreservation technology. Cryoprotective agents could be penetrating or non-penetrating depending on their permeability when crossing the membrane. Penetrating CPAs are a class of cryoprotectants that cross cell membranes, and they include ethylene glycol (EG), propylene glycol, DMSO, glycerol, and methanol. Non-penetrating CPAs are large molecules, usually polymers that inhibit ice growth via the same mechanisms as penetrating CPAs, but they do not enter cells [1]. Examples of non-penetrating cryoprotectants include sucrose, trehalose, and polyethylene glycol (PEG). Trehalose is a popular non-penetrating CPA, reported as less toxic and highly efficient in cryopreservation [11,12]. Additionally, a cocktail of non-penetrating (trehalose) and penetrating (glycerol) CPAs results in efficient post-cryopreservation recovery [11]. Non-penetrating CPAs are usually less toxic than their penetrating counterparts at the same concentration [13]. The toxicity of penetrating or permeating cryoprotectants were well-summarized in a recent review article [10].

Extreme environments are a less-explored niche, and these untapped ecosystems offer novel microbial metabolites with promising industrial applications [14]. Cold temperatures are widely distributed among the extreme environments on Earth. Cold environments have been successfully colonized by the microorganisms, which not only survive but actively metabolize at these freezing conditions. They have developed strategies to successfully overcome the negative effects of extremely low temperatures, including intracellular freezing [15]. The production of exopolysaccharide (EPS) is one of the many strategies used by cold-adapted microorganisms to cope with extremely low temperatures [16,17,18]. Exopolysaccharides are glycopolymers secreted by microorganisms in their surrounding environment [19]. These polymers are produced by a diverse group of microorganisms including bacteria, cyanobacteria, archaea, fungi, yeast, and microalgae. Sea ice and ocean particles in the Antarctic marine environment have abundant microbial EPS, and they could play a role in the survival and adaptation of microbial communities to cope with extreme temperatures and salinity [17,20,21].

Microorganisms are considered to be hidden wealth due to the enormous biotechnological potential they offer. Despite their huge potential, only few of bacterial EPSs have made their way into the global market, mainly because of their high production cost. Nevertheless, bioprospecting for novel EPSs with unique functional properties and improving their production cost can pave the way for their commercialization. The biodiversity and functionality of the Karakoram glaciers have not been fully explored and hence provide an opportunity for the discovery of novel microorganisms or their metabolites. Recently, Proteobacteria, Actinobacteria, and Bacteroidetes were found to be dominant in the Karakoram glaciers by using both culture and culture-independent methods [22]. A high proportion of bacterial isolates in this region produces antimicrobial compounds. *Pseudomonas* sp. strain BGI-2 was one of psychrotrophic bacteria isolated from the ice of Batura Glacier, Pakistan [23]. The BGI-2 strain was selected for further studies based on its maximum EPS production among the EPS-producing isolates. BGI-2 produces a high yield of cryoprotective EPSs at low temperatures. EPSs are categorized as stress molecules that assist microorganisms to cope with the extreme temperatures, high salinity, and desiccation [16,24]. Previously, the survivability of the EPS producer BGI-2 against a series of freeze–thaw cycles were compared to two other non-EPS-bacteria including *Rhodococcus* sp., BGI-11 isolated from the same environment, and a mesophilic *Escherichia coli*. The survivability of the EPS-producing BGI-2 strain was significantly higher than BGI-11 and *E. coli*. The EPS in our study also provided significant cryoprotection to another bacterium (*E. coli* K12), which was comparable to 20% glycerol [23]. This demonstrated the role of EPS in protecting cells from damage caused by freezing conditions and freeze–thaw events. Several studies have demonstrated the possible cryoprotective role of EPS in these cold and icy environments [16,25,26]. In another study, EPS from an Antarctic *Pseudomonas* species was reported for its cryoprotective role in the producer strain itself, as well as other bacteria [18].

EPSs from cyanobacteria and microalgae have been previously reported for their cryoprotective role in producer strains in extremely cold environments [24,27]. To the best of our knowledge, there has not been a single report of using a bacterial EPS for the cryopreservation of photosynthetic microorganisms including eukaryotic microalgae and prokaryotic cyanobacteria. The preservation of microalgae and cyanobacteria is important in basic research and industry applications. DMSO and methanol, the two most commonly used CPAs, are toxic to cells at room temperature [28]. These chemicals are also hazardous to humans when ingested, inhaled, or contacted through skin. We therefore used the possibility of employing a natural polymer (bacterial EPS) for the cryopreservation of the photosynthetic organisms.

In the current study, we tested the possibility of employing a bacterial EPS for the cryopreservation of photosynthetic microorganisms, including cyanobacteria and microalgae.

## 2. Material and Methods

### 2.1. EPS Yielded by Pseudomonas *sp.* BGI-2

*Pseudomonas* sp. BGI-2 is a bacterial strain isolated from the ice sample of Batura Glacier, Pakistan. BGI-2 is able to grow in a wide range of environmental conditions [23]. It can grow at temperatures of 4–35 °C, pH values of 5–11, and salt concentrations of 1–5%, and it utilize diverse sources of organic carbon. BGI-2 is known to produce a large quantity of EPSs, which contain rich sugar monomers like glucose, galactose, and glucosamine. A high copy number of EPS-producing genes was found in the BGI-2 genome, supporting its capability of high EPS production [29]. In this study, BGI-2 was grown at 15 °C and pH 6, in NaCl (10 g L^−1^), with glucose as the carbon source (100 g L^−1^), yeast extract as the nitrogen source (10 g L^−1^), and a glucose/yeast extract ratio of 10/1 to achieve the maximum EPS yield. Under these growth conditions, BGI-2 can yield 2 g L^−1^ of EPS.

### 2.2. Extraction of EPS

EPS was extracted following the procedure described by Ali et al. in 2020 [23]. Briefly, EPS produced by the BGI-2 bacterial strain was precipitated with ethanol and dried at room temperature. The crude EPS was further deproteinized with trichloroacetic acid (TCA), precipitated with ethanol, and dried. The EPS was re-dissolved in deionized water and dialyzed in a dialysis membrane (120 KDa molecular weight cutoff) to remove traces of TCA, salts, and low-molecular-weight molecules. The purified EPS was freeze-dried in a lyophilizer and used for cryopreservation by dissolving it in deionized water.

### 2.3. Cyanobacterial and Microalgal Strains Used for Cryopreservation Assay

Five strains, including 3 cyanobacteria and 2 microalgae, were used for this study. The cyanobacteria used in this study were *Synechococcus* sp. CB0101, *Synechococcus* sp. CBW1003, and *Microcystis aeruginosa* PCC7806. The microalgae chosen for this study were *Scenedesmus obliquus* HTB1 and *Chlorella vulgaris* UTEX 2714 (Appendix A). *Scenedesmus* sp. HTB1, *Chlorella vulgaris* UTEX 2714, and *Microcystis aeruginosa* PCC7806 were grown in a BG-11 medium [30], whereas the two *Synechococcus* strains (CB0101 and CBW1003) were grown in an SN medium [31] with a salinity of 15 ppt. *Microcystis aeruginosa* PCC7806 is a fresh water cyanobacterium which causes frequent algal blooms around the world. *Synechococcus* sp. CBW1003 is a picocyanobacteria isolated from the Chesapeake Bay during the winter season [32]. *Synechococcus* sp. CB0101 is another unicellular cyanobacterium isolated from the water of inner harbor Baltimore, Maryland [33]. This strain is a common inhabitant of the Chesapeake Bay, with a wide growth range for temperature, salinity, and nutrients. *Scenedesmus obliquus* HTB1 was isolated from the upper Chesapeake Bay (Back River) and has the ability to survive in high CO_2_ concentrations [34]. *Chlorella vulgaris* UTEX 2714 was purchased from the UTEX culture collection of algae at the University of Texas Austin, USA. 

### 2.4. Cryopreservation of Cyanobacteria and Microalgae

The cryoprotective effect of EPS for the cryopreservation of cyanobacteria and microalgae was determined by a method used previously with some modifications [35]. Three different concentrations of EPS (10%, 15%, and 20%) from a stock solution (20 mg/mL) were used for cryopreservation. The cryoprotective effect of EPS was compared to 5% (*v/v*) dimethyl sulfoxide (DMSO) and 10% (*v/v*) glycerol, which were used as controls. Glycerol and EPS were sterilized by autoclaving (121 °C for 15 min), whereas DMSO was sterilized using 0.2 µm filters. Each culture in their late log phase was transferred from a flask into falcon tubes and centrifuged at 10,000 rpm for 5 min at room temperature. The supernatant was discarded, and the tube with pellet was placed in ice. A fresh medium was added, and the pellet was resuspended. EPS was then added to a final concentration of 10%, 15%, or 20% (*v/v*), and 5% DMSO and 10% glycerol were added in a similar way. Cells were re-suspended by gentle vortexing and immediately transferred to cryovials (1 mL each). Cryovials were first incubated at 4 °C for 30 min and then stored in an ultra-low temperature freezer at −80 °C.

### 2.5. Growth Recovery

All 5 strains were checked for growth recovery after 5 days, 15 days, and 9 months of cryopreservation. For growth recovery, cryovials were taken out of the freezer and placed in a water bath at 35 °C for 5 min. Cultures were centrifuged at 8000 rpm for 3 min, and the supernatant was discarded to remove any CPAs. Cryopreserved cells were re-suspended in their respective media and transferred to 24-well plates (Appendix A). Cultures were incubated in the dark overnight in order to allow cells to recover under low light and under normal light at room temperature (21 °C) afterwards. Optical density at 750 nm was measured in a microplate reader (SpectraMax M5) to monitor growth.

### 2.6. Effect of Heat Sterilization on the Cryoprotective Activity of EPS

Three different concentrations of EPS were tested without autoclaving the stock to observe the impact of heat sterilization on the cryoprotective activity of the EPS.

## 3. Results

### 3.1. Growth Recovery of Cyanobacteria

#### 3.1.1. *Synechococcus* sp. CBW1003

The best growth recovery was obtained from EPS-preserved *Synechococcus* CBW1003 after five days of freezing at −80 °C. The maximum cell density, measured using optical density (OD) by proxy, was recorded at 20% EPS (OD 2.66), followed by 15% EPS (OD 2.13), 5% DMSO (OD 2.1), and 10% EPS (OD 1.98). A low cell growth was recorded in 10% glycerol (OD 1.44) and in the control (OD 0.95) after 18 days of incubation under normal shaking and light at room temperature (Figure 1a).

The maximum biomass recovery after 15 days of cryopreservation was also recorded at 20% EPS (OD 3.32), followed by 15% EPS (OD 3.27), 5% DMSO (OD 3.17), and 10% EPS (OD 3.01). Again, a low biomass recovery was recorded at 10% glycerol (OD 1.24) and in control (OD 1.52), as shown in Figure 1b.

The pattern of biomass recovery after 9 months of cryopreservation was similar to 5 and 15 days of cryopreservation. Growth after 9 months of cryopreservation showed the maximum recovery in 20% EPS (OD 2.32) and 5% DMSO (OD 2.34). A high cell density was also recorded at 15% EPS (OD 1.52) and 10% EPS (OD 1.52). The lowest biomass recovery was recorded at 10% glycerol (OD 0.591) and in the control (OD 0.76), as shown in Figure 1c.

#### 3.1.2. *Synechococcus* sp. CB0101

For *Synechococcus* sp. CB0101, the maximum recovery was recorded in 20% EPS (OD 2.34), followed by 15% EPS (OD 1.98) and 10% EPS (OD 1.74) after 5 days of cryopreservation. A poor recovery was recorded for 10% glycerol and in the control. No biomass was recovered at 5% DMSO even after 20 days of post-cryopreservation incubation under optimum conditions (Figure 2a).

A similar recovery pattern was recorded after 15 days of cryopreservation, with the maximum growth observed in cultures with EPS as the cryoprotective agent. The maximum biomass recovery was observed in the 20% EPS (OD 1.81), followed by 15% EPS (OD 1.45) and 10% EPS (OD 1.74). No biomass recovery was recorded at 5% DMSO, 10% glycerol, and in the control (Figure 2b).

Very similar results were obtained after the prolonged cryopreservation (9 months) of this strain. The maximum biomass recovery was recorded at 20% EPS (OD 2.26), followed by 15% EPS (OD 2.19) and 10% EPS (OD 1.30). Likewise, no recovery was observed in 5% DMSO, 10% glycerol, and the control after 26 days of incubation under optimum conditions (Figure 2c).

#### 3.1.3. *Microcystis aeruginosa* PCC7806

*Microcystis aeruginosa* PCC7806 demonstrated a good recovery in all treatments including the control. The maximum recovery after 5 days of cryopreservation was recorded at 20% EPS (OD 3.31), followed by 10% EPS (OD 3.30), 15% EPS (OD 3.26), the control (OD 3.18), and 5% DMSO (OD 2.89). Recovery was low at 10% glycerol (OD 2.84) compared to all other treatments (Figure 3a).

The recovery of this strain after 15 days of cryopreservation demonstrated a similar pattern, with the maximum recovery recorded at 15% EPS (OD 3.63), followed by 20% EPS (OD 3.58) and 15% EPS (OD 3.53). Biomass recovery in the control (OD 3.45) was better than 5% DMSO (OD 3.33) and 10% glycerol (OD 3.21) (Figure 3b).

Biomass recovery after 9 months of cryopreservation again demonstrated the maximum growth in the treatment with 20% EPS (OD 3.75), 15% EPS (OD 3.72), and 10% EPS (OD 3.60). The strain also recovered well in 5% DMSO (OD 3.54) and even in the control (OD 3.26). A poor recovery was observed in the 10% glycerol (Figure 3c).

### 3.2. Growth Recovery of Eukaryotic Microalgae

#### 3.2.1. *Scenedesmus* sp. HTB1

For the microalgal sp. *Scenedesmus* HTB1, the maximum growth recovery after 5 days of cryopreservation was observed at 5% DMSO (OD 2.15) followed by 10% EPS (OD 2.04), 15% EPS (OD 1.96) and 20% EPS (OD 1.78). Poor biomass recovery in 10% glycerol (OD 1.05) was observed. Viability and recovery were negligible in the control with no addition of any cryoprotective agent (Figure 4a). The optical density results presented here are over a period of 18 days of incubation under optimum conditions.

The maximum biomass recovery after 15 days of cryopreservation was again recorded at 5% DMSO (OD 1.87) and 10% EPS (OD 1.86), followed by 15% EPS (OD 1.80) and 20% EPS (OD 1.78). Low growth recovery was recorded at 10% (OD 1.35) and the control (OD 1.07) (Figure 4b). The pattern of biomass recovery after 9 months of cryopreservation was similar to 5 and 15 days of cryopreservation. Biomass recovery for HTB1 after 9 months of cryopreservation was the maximum at 10% EPS (OD 2.29) followed by 15% EPS (OD 2.19), 5% DMSO (OD 1.99) and 20% EPS (OD 1.92). The lowest growth recovery was again recorded at 10% glycerol and the control where optical density could not reach to 1 after 18 days of post-cryopreservation incubation (Figure 4c).

#### 3.2.2. *Chlorella vulgaris*

*Chlorella vulgaris* clearly demonstrated the maximum recovery (after 5 days of cryopreservation) in presence of 5% DMSO (OD 3.32) as a CPA followed by 10% glycerol (OD 1.95) and the control (OD 1.67) after 18 days of incubation at optimum conditions. A poor recovery was observed for all concentrations of EPS used (Figure 5a). The optical density results presented here were over a period of 18 days of incubation under optimum conditions.

Fifteen days of cryopreservation results again demonstrated 5% DMSO (OD 3.26) as the choice of cryoprotective agent for this strain. Recovery was better in the control than the treatments containing EPS as the cryoprotective agent (Figure 5b), and 5% DMSO worked well for the longer preservation of this strain.

After 9 months of cryopreservation, the maximum biomass was recovered from 5% DMSO (OD 2.71), followed by 10% EPS (OD 2.13) and the control (OD 1.94). Recovery was low at all other concentration of EPS and glycerol (Figure 5c). The optical density results presented here were over a period of 18 days of incubation at the optimum conditions.

### 3.3. Effect of Heat Sterilization on the Cryoprotective Activity of the EPS

The autoclaving of EPS had no major effects on its cryoprotective activity for majority of the strains. However, in case of *Synechococcus* sp. CB0101, there was clear difference in biomass recovery in the treatments with autoclaved and non-autoclaved EPSs. As discussed above, the CB0101 strain demonstrated the maximum biomass recovery in the EPS, while no recovery was recorded at 5% DMSO and 10% glycerol. The maximum biomass recovery was recorded at 20% non-autoclaved EPS (OD 2.26). Comparatively, the biomass recovery was low in the presence of autoclaved EPS as the cryoprotective agent (Figure 6).

## 4. Discussion

Overall, EPS worked well as the cryoprotective agent for strains including *Scenedesmus* sp. HTB1, *Synechococcus* sp. CBW1003, *Synechococcus* sp. CB0101, and *Microcystis* sp. 7806. For all these strains, growth recovery was better than 5% DMSO, a common CPA used for the cryopreservation of various cells. A poor recovery was recorded when 10% glycerol was used as the cryoprotective agent. There are conflicting reports regarding the performance and choice of CPAs for the cryopreservation of photosynthetic organisms. Gaget et al. (2017) used different concentrations of methanol, DMSO, and glycerol for the cryopreservation of 196 cyanobacterial isolates and found 5% DMSO to be the preferable choice of CPA for most of the strains [35]. In our study, 5% DMSO (except for the CB0101 strain), along with the EPS, also worked well for most of the strains. In another study, Esteves-Ferreira et al. (2013) used five microalgae and cyanobacterial strains for cryopreservation and found 10% glycerol to be the most efficient CPA [36]. Some studies have demonstrated the effectiveness of CPAs when used in combination rather than alone. Nakanishi et al. (2012) demonstrated the 50% survivability of the four microalgal strains when a combination of CPAs (5% DMSO, 5% ethylene glycol, and 5% proline) was used [37]. According to their findings, little or no cryoprotection was observed when these CPAs were used alone. Aray-Andrade et al. (2018) found DMSO–sucrose and glycerol to be effective cryoprotective agents while working with the cryopreservation of two *Chlorella* and one *Scenedesmus* species [38].

In *Synechococcus* sp. CB0101, EPS was the only cryoprotective agent where growth was recovered after 9 months of cryopreservation, whereas no recovery was observed with DMSO and glycerol. Interestingly, growth recovery was significantly higher in the non-autoclaved EPS than in the autoclaved EPS. This pattern of growth recovery was recorded at all three durations of cryopreservation (5 days, 15 days, and 9 months). For the CB0101 strain, a higher concentration of EPS (20%) worked well, whereas 5% DMSO exhibited a toxic effect. Though 5% DMSO is the preferable CPA, it has previously been reported for its cytotoxic effect to susceptible cyanobacterial strains {36]. Other studies have also reported the toxicity of DMSO [39,40]. Penetrating CPAs that cross cell membranes, namely EG, propylene glycol, DMSO, glycerol, and methanol, have been reported to have cytotoxic activity [10]. For the successful post-thaw survival rate of microalgae and cyanobacteria, the type and concentration of cryoprotectants are crucial. In our study, EPS worked well for all the strains except for *Chlorella vulgaris*. Likewise, 5% DMSO performed well as a CPA for most of the strains except for *Synechococcus* sp. CB0101. Similarly, the same cryoprotectant with a different concentration had a different growth recovery. It is therefore critical to test the type and concentration of cryoprotectants for each strain prior to long-term cryopreservation. The choice of CPA, viability, and biomass recovery are all very much strain-dependent [41].

The BGI-2 producer strain is a psychrotrophic bacterium capable of growing at low temperatures. This strain has a high yield of EPS (2 g L^−1^) at 15 °C, which negates the expensive heating steps required for working with mesophilic strains. Working at low temperatures conserves energy, minimizes contamination with other microorganisms, and minimizes undesirable chemical reactions. Despite the enormous biotechnological potential, only a handful of bacterial EPSs have been successfully commercialized. The major hindrance is the cost of production, which can be overcome by successfully employing a number of measures including the use of an inexpensive substrate, fermentation optimization to improve product yield, the improvement of the producer strain through mutagenesis, genetic and metabolic manipulations to enhance the productivity, and the improvement of downstream processing, which involves extraction and purification.

## 5. Conclusions

Many laboratories worldwide use sub-culturing as the primary method for cyanobacterial and microalgal culture maintenance despite inherent disadvantages. This study serves as the first step towards successful use of EPS for cryopreservation photosynthetic microorganisms including cyanobacterium and eukaryotic microalgae. The concentration of EPS used as a CPA is critical factor that is dependent on the sensitivity of a particular strain. Most of the strains demonstrated good recovery and viability at the higher EPS concentrations used. Therefore, more tests with increased EPS concentrations will further improve growth recovery. More research can improve methods and standard operating protocols for this natural polymer to replace the existing toxic chemicals in use today as cryoprotective agents.

## Figures and Tables

**Figure 1 microorganisms-09-00395-f001:**
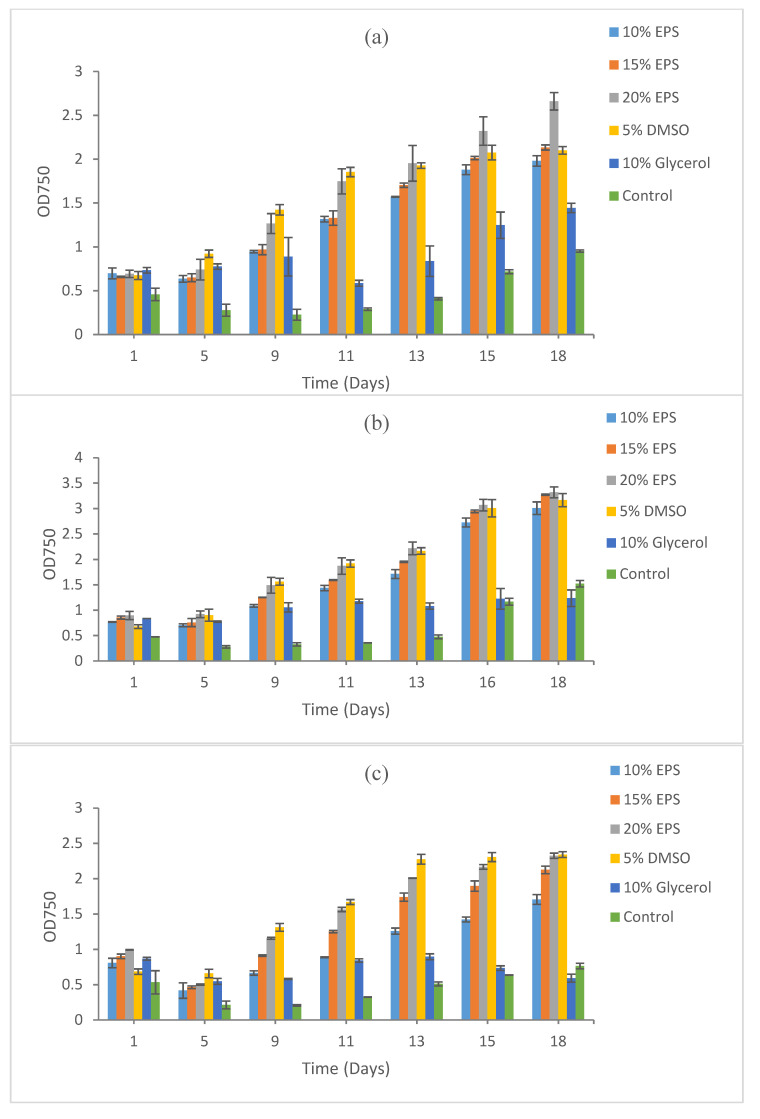
Growth recovery for *Synechococcus* sp. CBW1003 after cryopreservation for (**a**) 5 days, (**b**) 15 days, and (**c**) 9 months. Duplicate samples were measured. The images of cultures at the end points refer to Appendix A. DMSO: dimethyl sulfoxide; EPS: exopolysaccharide.

**Figure 2 microorganisms-09-00395-f002:**
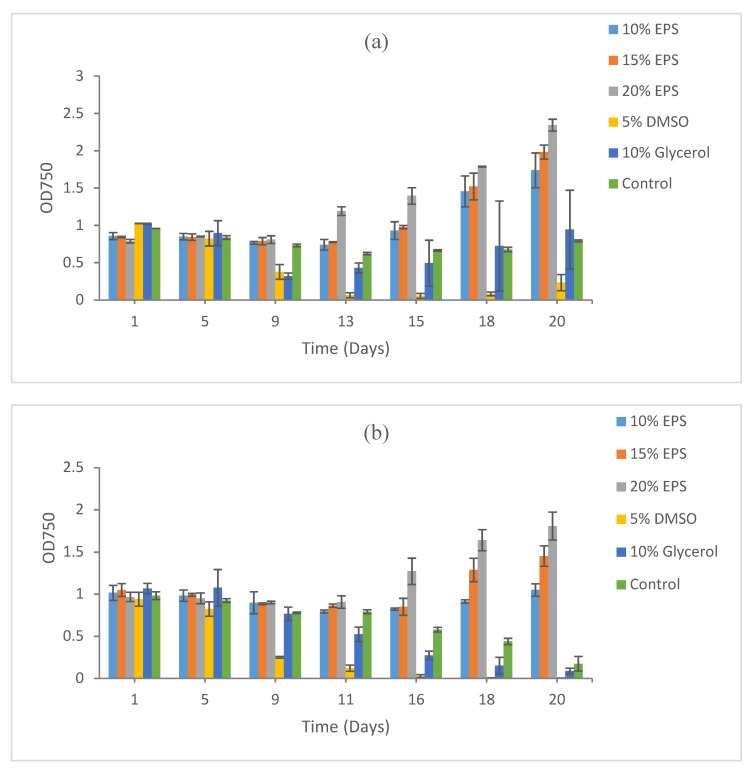
Growth recovery for *Synechococcus* sp. CB0101 after cryopreservation for (**a**) 5 days, (**b**) 15 days, and (**c**) 9 months. Duplicate samples were measured. The images of cultures at the end points refer to Appendix A.

**Figure 3 microorganisms-09-00395-f003:**
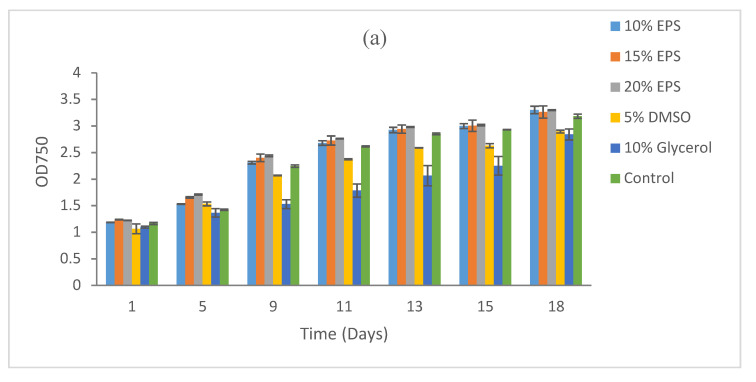
Growth recovery for *Microcystis aeruginosa* PCC7806 after cryopreservation for (**a**) 5 days, (**b**) 15 days, and (**c**) 9 months. Duplicate samples were measured. The images of cultures at the end points refer to Appendix A.

**Figure 4 microorganisms-09-00395-f004:**
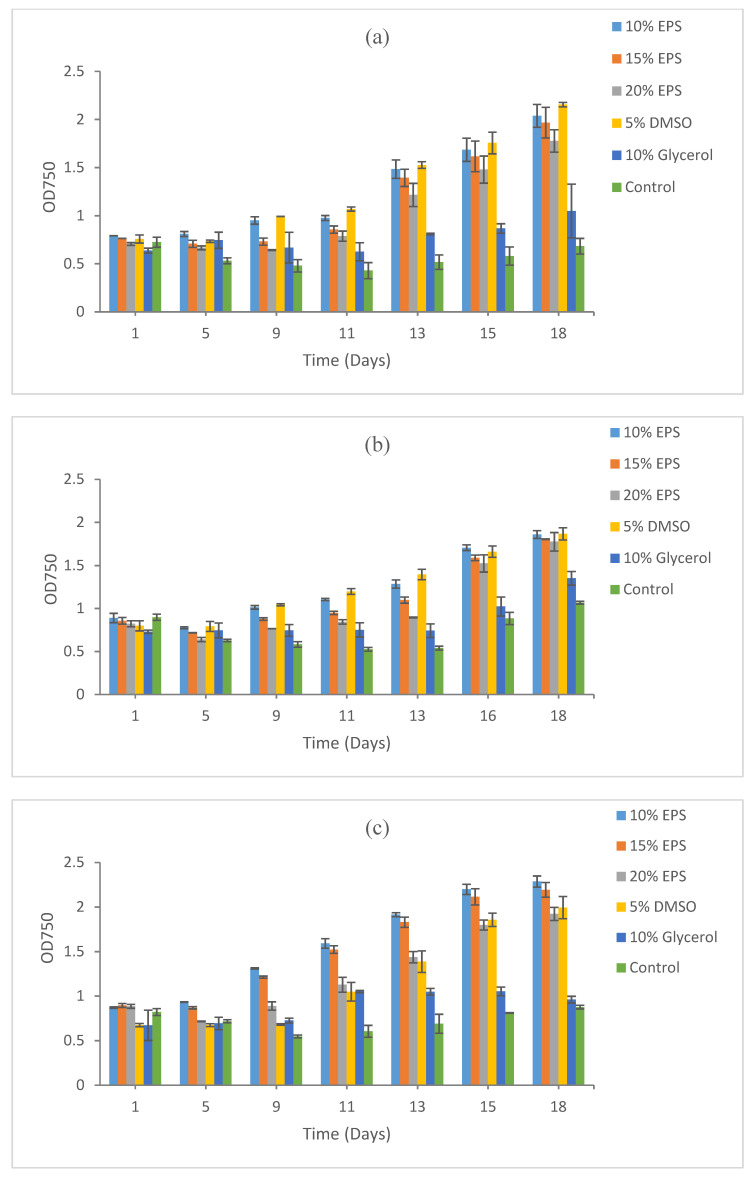
Growth recovery for *Scenedesmus* sp. HTB1 after cryopreservation for (**a**) 5 days, (**b**) 15 days, and (**c**) 9 months. The images of cultures at the end points refer to Appendix A.

**Figure 5 microorganisms-09-00395-f005:**
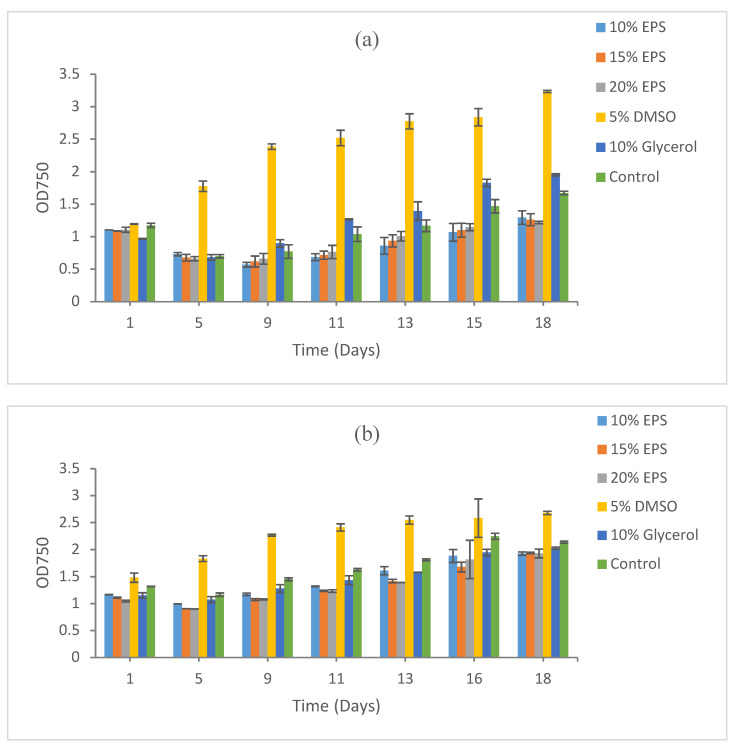
Growth recovery for *Chlorella vulgaris* after cryopreservation for (**a**) 5 days, (**b**) 15 days, and (**c**) 9 months. Duplicate samples were measured. The images of cultures at the end points refer to Appendix A.

**Figure 6 microorganisms-09-00395-f006:**
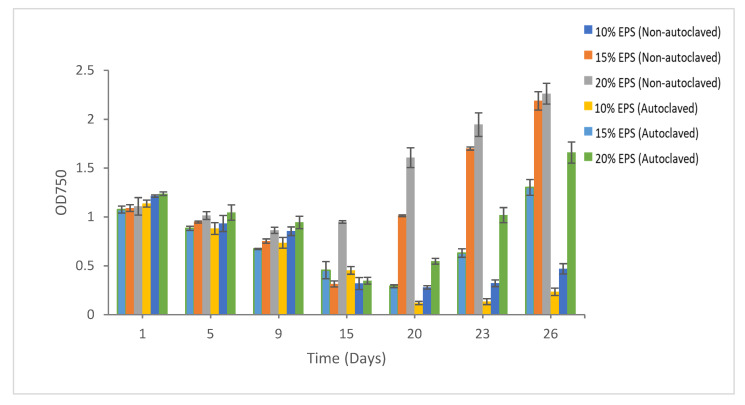
Growth recovery for *Synechococcus* sp. CB0101 after 9 months of cryopreservation in autoclaved and non-autoclaved EPSs as the cryoprotective agents. Duplicate samples were measured.

## Data Availability

MDPI Research Data Policies.

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
