# Peer review of "Cryopreservation of Cyanobacteria and Eukaryotic Microalgae Using Exopolysaccharide Extracted from a Glacier Bacterium"

_microorganisms, 2021, doi:10.3390/microorganisms9020395_

Round 1
Reviewer 1 Report
The manuscript by Ali et al. deals with the cryopreservation of selected cyanobacterial and microalgal strains by using EPSs previously obtained from a glacier isolate (Pseudomonas sp. BGI-2). The paper is interesting for an applicative purpose. However, in my opinion, the manuscript needs to be rearranged in some parts to better valorize the work. I suggest a major revision. Specific comments are listed below:
- line 9, Exopolysaccharides (EPSs) have been.... Modify accordingly
- lines 20-21, ....from a psychrophilic bacterium... What bacterium? Pseudomonas sp. BGI-2 from the ice of Batura glacier must be cited also in the abstract. At line 99, you report BGI-2 as psychrotrophic, whereas it is psychrophilic at line 20. These are two different classifications. Please, modify accordingly.
- line 34, typically at -80 °C
- line 35 and others, citations should be numbered, I guess
- line 37, sub-culturing
- lines 82-83, exopolysaccharides (EPSs)... Please, be carefully when addressing to EPS or EPSs in the entire manuscript.
- lines 88-89, not only in Antartica as a cold environment. Please, modify.
- lines 96-98, please, better introduce this topic.
- lines 98-106, please reduce this text only introducing the strain BGI-2. Then, move it at the beginning of methods. In such paragraph, main results for BGI-2 (Ali et al., 2019, 2020) should be summarized in a table. This will be useful to the readers. Do you have a chemical characterization of the EPSs used in this study? This aspect should be stressed to valorize this exoproduct. What about the production yield? What about the laboratory conditions to produce EPSs?
- lines 110-118, please rewrite by highlighting the main research question.
- lines 114-116, This sentence is a result, mores uitable for the abstract. Please, remove from the introduction.
- line 168, please rephrase.
- Results, why did you alternate cyanobacteria and microalgae in this section? This make confusion. I suggest to group results for cyanobacteria in a single paragraph and related sub-paragraphs. The same should be done for microalgae.
- Figures 1 to 6, OD750 instead of optical density
- lines 284-298, such information should be given in the introduction to valorize the strain, the EPSs it produces and the EPS application as cryoprotective agent for subzero preservation od cyanobacteria and microalgae.
- line 301, EPS-BGI-2, this is the first time you mention EPSs from BGI-2 in the manuscript in this form. Please, modify accordingly
- lines 299-311, again this text should be moved to the introduction. Discussions should refer mainly to results obtained in this study, also comparing with previous studies.
- line 315, using only CPA is sufficient
- lines 348-357, this concept has been already reported above in the manuscript. Please, remove.
- line 359, ...a high yield of EPS a... How much? This in a focal point of your study. Is BGI-2 so efficient in EPS production to be applied at a larger scale?
- Figure S1, please improve resolution of circles.
Author Response
We really appreciate your careful editing on our manuscript, and have incorporated changes you suggested. The manuscript flows much better with these changes.
We will format the references when you are satisfied with these changes. It is kind of difficult to do more changes in the current vision (too many changes already).
I attached our responses in a separate file, in which the text in blue color highlights our point-by-point responses.
Thank you again,

Reviewer 2 Report
The paper is clear and well written.
The topic of the paper is appropriate for the Journal.
The text of the paper was written correctly in terms of stylistically, punctuation and terminology.
The paper was correctly edited and graphically developed at a good level.
The literature was chosen correctly and fully used in the paper.
I do not any shortcuts.
Correct the citation of literature in the text. There should be numbers.
The paper deserves a positive assessment because it is current and interesting form both a cognitive and practical point of view.
Author Response
Thank you for your positive feedback on our work. We will change the citation format according to the journal requirement.
Round 2
Reviewer 1 Report
The authors have addressed all my comments and suggestions. I am happy to have been helpful
Author Response
Thank you!